# Efficient Decolorization of Azo Dye Orange II in a UV-Fe$^{3+}$-PMS-Oxalate System

Yajie Wang [1], Xin Dong [1], Chengfeng Liu [1], Peng Cheng [2] and Gilles Mailhot [2,*]

1 School of Eco-Environmental Engineering, Guizhou Minzu University, Guiyang 550025, China
2 Institut de Chimie de Clermont Ferrand (ICCF) UMR 6296, Université Clermont Auvergne, CNRS, Clermont Auvergne INP, BP 80026, F-63171 Clermont-Ferrand, France
* Correspondence: gilles.mailhot@uca.fr

**Abstract:** The decolorization of azo dye Orange II using a UVA-Fe$^{3+}$-PMS-oxalate system was studied. A series of experiments was performed to investigate the effects of several variables, including the pH, PMS dosage, Fe$^{3+}$ concentration, oxalate concentration, and coexisting anions. The results revealed that a lower pH facilitated the decolorization, and relatively high decolorization efficiency (97.5%) could be achieved within 5 min at pH 3.0. The electron paramagnetic resonance (ESR) and radical quenching experiments revealed that SO$_4^{\bullet-}$ played a crucial role in the decolorization of Orange II (85.8%), $^\bullet$OH was of secondary importance (9%), and $^1$O$_2$ made a small contribution to the decolorization (5.2%). Furthermore, the formation of $^\bullet$OH in the experimental system strongly depended on HO$_2^\bullet$/O$_2^{\bullet-}$. These reactive oxidants were able to directly attack the azo bond of the luminescent group in Orange II and initiate the decolorization process. The efficient UVA-Fe$^{3+}$-PMS-oxalate system showed great application potential in the treatment of wastewater contaminated by azo dyes.

**Keywords:** PMS activation; Orange II; decolorization; UVA irradiation; Fe$^{3+}$-oxalate complexes

## 1. Introduction

Wastewater discharged from tanneries, textile, food, and paper production industries is one of the most critical environmental problems nowadays [1]. These have negative impacts on the health of humans, animals, and the environment. Nearly half of the global productions of synthetic textile dyes are classified as azo compounds. Azo dyes are highly carcinogenic due to amine and benzidine emissions. Azo group proliferation is difficult to degrade by traditional biological treatment methods due to their complex structures and stabilities. Therefore, it is of great environmental concern to develop effective and feasible treatment technologies to treat azo dye contamination.

Advanced oxidation processes (AOPs) have been gaining more attention for the detoxification of water contaminated with azo dyes. Some strong oxidative species (e.g., $^\bullet$OH, HO$_2^\bullet$/O$_2^{\bullet-}$) can be generated in AOPs and are very effective in dye bleaching and even mineralization [2]. AOPs have also become attractive technologies due to the usage of economical reagents (e.g., hydrogen peroxide, peroxydisulfate (PDS), and peroxymonosulfate (PMS)) and the potential to mineralize contaminants through radical generations [3]. In recent years, sulfate radical anion (SO$_4^{\bullet-}$)-based oxidation has gained attention for the efficient degradation of non-biodegradable contaminants in aquatic environments [3–5]. Furthermore, SO$_4^{\bullet-}$, with a high standard redox potential (2.5–3.1 V), higher than that of the hydroxyl radical ($^\bullet$OH, 1.8–2.7 V, depending on the pH), transformed the contaminants into simpler or more degradable products, or even mineralized the contaminants in wastewater [6]. SO$_4^{\bullet-}$ can be readily generated from the decomposition of PDS and PMS by thermal activation, transition metal (i.e., Fe$^{2+}$, Co$^{2+}$), and irradiation [3]. The use of PMS and UV irradiation at 254 nm was employed for the generation of sulfate radicals to

degrade many organic contaminants [7]. However, low wall-plug efficiency and high costs of UV irradiation at 254 nm restrict its application. As one of the common metal activators, ferrous ion ($Fe^{2+}$) has been widely used for PMS activation due to its cost-effectiveness and environmental non-toxicity [8]. The photo-transformation of $Fe^{3+}$ into $Fe^{2+}$ by UVA irradiation is the source of $Fe^{2+}$ to activate PMS, and $Fe^{3+}$ could be regenerated from the activation of PMS by $Fe^{2+}$. These two steps may form a closed-loop process until the complete consumption of PMS. Therefore, a treatment process combining UVA, $Fe^{3+}$, and PMS might provide an appreciable degradation of contaminants.

However, the major obstacle to this process is the slow rate of $Fe^{3+}$ transformation back to $Fe^{2+}$. Employing different chelates enhanced the oxidation ability and reduced the formation of iron sludge by accelerating the regeneration of $Fe^{2+}$ [9]. Natural polyphenols [10], 3,4,5-trihydroxybenzoic acid [8], epigallocatechin-3-gallate [11], ethylenediaminetetraacetic acid [12], and oxalic acid [13] have been found to accelerate the $Fe^{3+}$/ $Fe^{2+}$ cycling in $SO_4^{\bullet-}$-based AOPs. It was demonstrated that oxalate is an effective chelating agent for $Fe^{3+}$ [10], and oxalate has a simple structure with a bidentate ligand, which contains two carboxyl groups and two pairs of electrons [13]. The ligands of oxalate could decline the standard redox potential of $Fe^{3+}/Fe^{2+}$ coupling [14], affecting the activity of iron complexes ($Fe^{3+}/Fe^{2+}$ = 0.77 V, $Fe^{2+}/Fe^{3+}$-oxalate = 0.002 V). The addition of oxalate facilitates the redox ability of $Fe^{2+}$ and improves the reactions between the iron and the oxidant. In addition, it was reported that oxalic acid and its ionophores have a lower reactivity [15] so it cannot react with the target contaminants in a significantly competitive manner, thus avoiding significant depletion of the active species. Few papers have reported the introduction of a chelating agent into the UVA-$Fe^{3+}$-PMS system.

In this study, Orange II was used as the model contaminant of an azo dye, whose decolorization was studied in a UVA-$Fe^{3+}$-PMS-oxalate system. The effects of the pH, initial concentrations of PMS, oxalate, $Fe^{3+}$, and coexisting anions on the decolorization process were investigated. Additionally, the active oxidants involved and the mechanisms by which oxalate promoted the catalytic performance of the UVA-$Fe^{3+}$-PMS process were elucidated. Our work may provide insight into the operational parameters for the treatment of azo dye using the UVA-$Fe^{3+}$-PMS-oxalate system.

## 2. Materials and Methods

### 2.1. Chemicals and Reagents

Orange II (4-(2-hydroxy-1-naphthylazo) benzenesulfonic acid, >85.0%) was obtained from Shanghai Macklin Biochemical Co., Ltd. (Shanghai, China). Potassium monopersulfate (OXONE®, $KHSO_5 \cdot 0.5KHSO_4 \cdot 0.5K_2SO_4$, ≥99.0%), potassium persulfate (PDS, $K_2S_2O_8$, ≥99.0%), iron(III) perchlorate hydrated ($Cl_3FeO_{12} \cdot 12H_2O$, ≥99.0%), and ferrozine ($C_{20}H_{13}N_4NaO_6S_2$, ≥97.0%) were purchased from Sigma-Aldrich (USA). Benzoquinone ($C_6H_4O_2$, 99.0%), furfuryl alcohol ($C_5H_6O_2$, 98.0%), sodium propionate ($C_3H_5NaO_2$), sodium pyruvate ($C_3H_3NaO_3$), sodium gluconate ($C_6H_{11}NaO_7$), sodium succinate ($C_4H_4Na_2O_4$), and glutaric acid ($C_5H_8O_4$) were obtained from Aladdin Co., Ltd. (China). Tert-butyl alcohol (TBA), isopropanol (IPA), sodium oxalate ($Na_2C_2O_4$), and sodium citrate ($C_6H_5Na_3O_7$) were purchased from Sinopharm Chemical Reagent Co., Ltd. (China). Hydrogen peroxide ($H_2O_2$, 35%) was obtained from Chongqing Jiangchuan Chemicals, China. Ultrapure water (18 MΩ cm) was used in all experiments.

### 2.2. Experimental Procedures

Batch experiments were conducted in glass bottles (250 mL) under magnetic stirring (400 rpm) at a constant water bath temperature (293 K). The desired amounts of Orange II, $Fe^{3+}$, oxalate, and PMS were added to the solution, and the solution pH was adjusted with a $HClO_4$ or NaOH solution. Reactions were initiated after the UVA light was turned on. Each sample was collected at specific time intervals, and residual Orange II in the mixture was analyzed by a spectrometric method at a wavelength of 484 nm using a UV795 UV-vis spectrophotometer (Yoke China).

Quenching experiments were conducted by adding specific concentrations of IPA, TBA, CHCl$_3$, and FFA to the solution, which was prepared using the same procedure for the solutions of PMS, Fe$^{3+}$, oxalate, and Orange II to identify the contributions of different reactive oxidants to Orange II decolorization in the experimental system. Due to the high volatility of CHCl$_3$, the top of the flask was sealed with a rubber stopper in the O$_2^{\bullet-}$ scavenging experiments, and the magnetic stirring method was replaced by bubbling air to keep the suspension well-mixed.

### 2.3. Analytical Methods

The concentrations of ferrous ions in the solutions (Fe$^{2+}$) were determined using ferrozine and a spectrophotometer at a wavelength of 562 nm ($\varepsilon_{562}$ = 27,900 M cm$^{-1}$) [16]. The total aqueous iron (Fe(T)) was determined by adding ascorbic acid to reduce Fe(III) to Fe(II). The presence of SO$_4^{\bullet-}$, $^{\bullet}$OH, $^1$O$_2$, and O$_2^{\bullet-}$ was detected by electron spin resonance (ESR) using a JEOL JES-FA300 ESR spectrometer with 60 mM 5,5-dimethyl-1-pyrroline-N-oxide (DMPO) as a spin trapping agent for $^{\bullet}$OH and SO$_4^{\bullet-}$ and for HO$_2^{\bullet}$/O$_2^{\bullet-}$ in water; 2,2,6,6-tetramethyl-4-piperidinol (TEMP) was used for $^1$O$_2$.

To compare the reaction rate constants under different conditions, the pseudo-first-order model was used to fit the kinetic data. The pseudo-first-order equation can be expressed as Equation [17]:

$$\ln(C/C_0) = -k_{\mathrm{obs}}\, t,$$

where $C_0$ and $C$ are the concentrations at an initial time and any reaction time $t$, respectively. $k_{\mathrm{obs}}$ (min$^{-1}$) was calculated from the slope of the plots of $\ln(C/C_0)$ vs. $t$.

## 3. Results and Discussion

### 3.1. Decolorization of Orange II in Different Peroxides Systems (PDS, PMS, and H$_2$O$_2$)

The decolorization of Orange II in the different systems is shown in Figure 1. In the 30 min reaction, the decolorization efficiencies in the UVA-Fe$^{3+}$-H$_2$O$_2$-oxalate system, UVA-Fe$^{3+}$-PDS-oxalate system, and UVA-Fe$^{3+}$-PMS-oxalate system were 31.61%, 73.9%, and 95.04%, respectively. The different decolorization efficiencies of Orange II in the different peroxides systems could be explained by the generation of sulfate radical (SO$_4^{\bullet-}$) and low bond energy of PMS. A part of these peroxides dissociated after UVA irradiation, as illustrated in the following reactions [6]; however, most of them reacted with the generated Fe$^{2+}$ to produce strong oxidants.

$$\mathrm{H_2O_2} + h\upsilon \rightarrow 2^{\bullet}\mathrm{OH}, \tag{1}$$

$$\mathrm{S_2O_8}^{2-} + h\upsilon \rightarrow 2\mathrm{SO_4}^{\bullet-}, \tag{2}$$

$$\mathrm{HSO_5}^- + h\upsilon \rightarrow \mathrm{SO_4}^{\bullet-} + {}^{\bullet}\mathrm{OH}, \tag{3}$$

In the H$_2$O$_2$ system, a photo-Fenton system was formed, and its main oxidant was $^{\bullet}$OH. In the PMS and PDS systems, the main oxidant was SO$_4^{\bullet-}$. The standard redox potential of SO$_4^{\bullet-}$ (2.5–3.1 V) is higher than that of $^{\bullet}$OH (1.8–2.7 V), which may have caused the UV/PMS system to have a stronger oxidizing capacity. It has been found that PMS is more likely to exhibit heterolytic cleavage due to its non-symmetrical structure [1,18]. Therefore, it might be easier for PMS to generate homolytic cleavage compared to PDS. The S-S bond in PDS is stronger than the S-O bond in PMS, so it is more difficult to break the S-S bond than the S-O bond. Previous studies have also found that PMS showed better performance in metallic activation compared to PS or H$_2$O$_2$ [19,20]. In this study, the UVA-Fe$^{3+}$-PMS-oxalate system demonstrated better kinetic performances over the UV/H$_2$O$_2$ and UV/PDS systems for Orange II decolorization. Therefore, PMS was chosen as the source of SO$_4^{\bullet-}$, and the factors influencing the decolorization of azo dyes mechanisms in the UVA-Fe$^{3+}$-PMS-oxalate system were investigated. The absorption spectra of Fe$^{3+}$, PMS, oxalate, Fe$^{3+}$-oxalate complexes, and Orange II are shown in Figure S1.

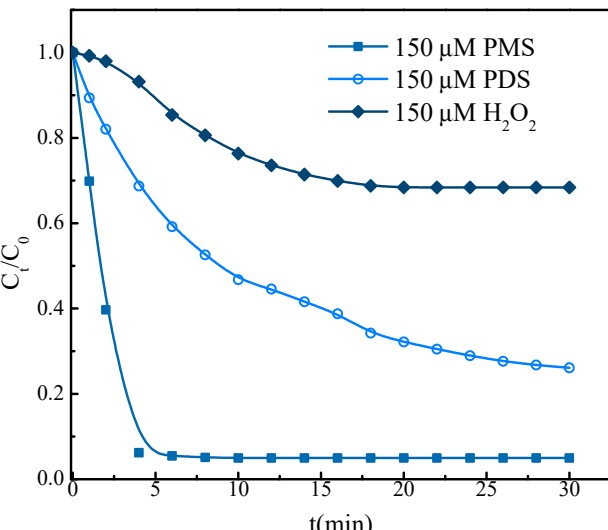

**Figure 1.** The decolorization of Orange II in different peroxides systems ($[Fe^{3+}]_0$ = 100 μM, $[Orange\ II]_0$ = 50 μM, $[H_2O_2]_0$ = $[PDS]_0$ = $[PMS]_0$ = 150 μM, $[oxalate]_0$ = 80 μM, under UVA irradiation, $\lambda_{irr.}$ = 365 nm, pH = 3).

### 3.2. Effects of Carboxylates on the Decolorization of Orange II in UVA-$Fe^{3+}$-PMS System

In the natural environment, small molecules of carboxylic acids can combine with $Fe^{3+}$ to form $Fe^{3+}$-polycarboxylate complexes. It is well known that $Fe^{3+}$-polycarboxylate complexes undergo ligand-to-metal charge transfer (LMCT) upon light irradiation, resulting in a series of reactive oxidative species [21,22]. Seven different carboxylates (oxalate, pyruvate, propionate, succinate, glutarate, citrate, and gluconate) were employed to investigate the effects of different carboxylates on the decolorization efficiency of Orange II.

As shown in Figure 2, the addition of oxalate, pyruvate, citrate, and gluconate significantly promoted the decolorization of Orange II compared to the system without carboxylate, indicating that these carboxylates complexed with $Fe^{3+}$ and produced photoreactive complexes, accelerating the $Fe^{3+}/Fe^{2+}$ cycle. The addition of glutarate slightly promoted the decolorization of Orange II. The addition of succinate and propionate had a slight inhibitory effect on the decolorization of Orange II, with the lowest value of $k_{obs}$ (0.037 min$^{-1}$). The complexation of glutarate with $Fe^{3+}$ formed a low photo-reactive complex. The inhibition effects of succinate and propionate may be ascribed to two factors: firstly, $Fe^{3+}$ was still present as Fe(III)-OH in the system, probably due to the low complexation constants of succinate and propionate acid with $Fe^{3+}$ [23]; secondly, succinate or propionate would compete with Orange II for the reactive oxidants produced by the photolysis of $Fe^{3+}$.

Of the seven carboxylates, the presence of oxalate imposed the most significant enhancement to the decolorization of Orange II, and the decolorization efficiency of Orange II was 97.5% within 15 min. The initial decolorization rate was the fastest in the presence of oxalate, with the highest $k_{obs}$ of 0.515 min$^{-1}$. Ferrioxalate is an $Fe^{3+}$-polycarboxylate complex with high photoactivity [22]. Under irradiation, ferrioxalate generates carbon-centered radicals and $Fe^{2+}$ (Reactions 4 and 5). As for carbon-centered radicals, they transform into $O_2^{\bullet-}$ in the presence of dissolved oxygen (Reaction 6) [24,25].

$$[Fe^{III}(C_2O_4)_3]^{3-} + h\nu \rightarrow [Fe^{II}(C_2O_4)_2]^{2-} + C_2O_4^{\bullet-}, \tag{4}$$

$$[Fe^{II}(C_2O_4)_2]^{2-} + h\nu \rightarrow Fe^{II}(C_2O_4) + C_2O_4^{\bullet-}, \tag{5}$$

$$C_2O_4^{\bullet-} + O_2 \rightarrow O_2^{\bullet-} + 2CO_2, \tag{6}$$

More importantly, the generated $Fe^{2+}$ activated the PMS to generate strong oxidants to decolorize the Orange II. The production of $Fe^{2+}$ was monitored in the experimental system; it increased from 0.0 to 42.24 μM within 10 min (Table S1). Ferrioxalate with a

high quantum yield acted as a source of $Fe^{2+}$ under UVA irradiation and supported the subsequent activation of PMS. Concerning the decolorization efficiency and rate constant of Orange II, oxalate was used as the chelating agent in this study.

Additionally, the decolorization of Orange II in the $Fe^{3+}$-PMS-oxalate system without irradiation was investigated (Figure S2). No decolorization was observed in the dark, implying that UVA irradiation initiated the process of PMS activation, and then Orange II decolorization. Furthermore, Orange II decolorization under different irradiation wavelengths was also studied, and the results are presented in Figure S3. Under a wavelength of 395 nm, no decolorization was found. The decolorization process successfully occurred under 254 nm and simulated sunlight irradiation.

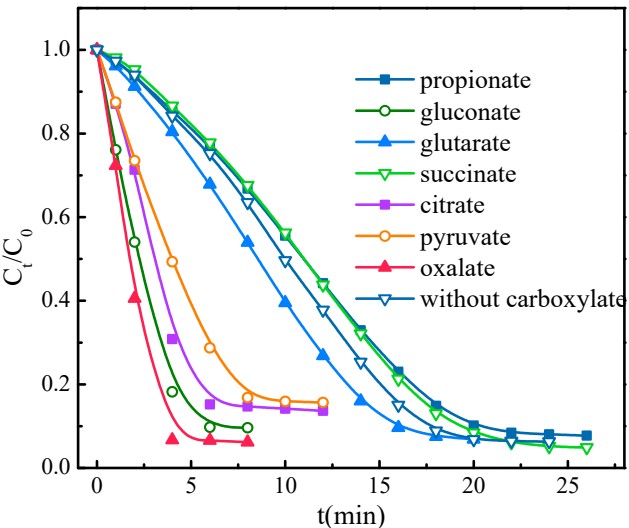

**Figure 2.** The effects of different carboxylates on the decolorization of Orange ($[Fe^{3+}]_0 = 100$ μM, $[Orange\ II]_0 = 50$ μM, $[PMS]_0 = 150$ μM, $[carboxylates]_0 = 80$ μM, under UVA irradiation, $\lambda_{irr.} = 365$ nm, pH = 3).

### 3.3. Effect of the Initial pH on the Decolorization of Orange II in UVA-$Fe^{3+}$-PMS System

The pH has a profound effect on the speciation of $Fe^{3+}$-carboxylate complexes and the generation of radicals in $SO_4^{\bullet-}$-based AOPs. In order to investigate the effects of the initial pH on the decolorization of Orange II in the UVA-$Fe^{3+}$-PMS-oxalate system, experiments were performed at different solution pH conditions (1.0, 2.0, 3.0, 4.0, 5.0, 6.0, 7.0, 8.0, and 9.0).

The effects of the pH on the decolorization of Orange II are shown in Figure 3. As the pH increased from 1.0 to 3.0, the decolorization efficiencies increased from 77.4% to 97.5%. When the pH increased from 3.0 to 5.0, the decolorization of Orange II obviously decreased. With the further increasing pH from 6.0 to 9.0, no decolorization of Orange II was observed in the experimental system. The highest decolorization efficiency was achieved at pH 3.0, with a decolorization efficiency of 97.5% within 10 min.

At pH 3.0, the complexes of $Fe^{3+}$ and oxalate are mainly present in the forms of $Fe(C_2O_4)_2^-$ and $Fe(C_2O_4)_3^{3-}$, which have high photoactivity and can rapidly photolyze to provide $Fe^{2+}$ and then activate the PMS [25,26]. With increasing pH levels (4.0–5.0), $Fe^{3+}$-oxalate complexes are mainly present as $Fe(C_2O_4)_2^+$ and $Fe(C_2O_4)^+$, which show relatively lower photoactivity [27]. At pH 6.0 and above, most $Fe^{3+}$ is present mainly as $Fe(OH)_3$, which is not photoreactive [28]. The redox cycle of $Fe^{3+}/Fe^{2+}$ was completely halted; therefore, the decolorization of Orange II did not take place under such pH conditions. When the pH was lower than 3.0, the percentage of free $Fe^{3+}$ increased with the decrease in pH, which was less photoactive, resulting in a slower $Fe^{2+}$ production rate and a lower decolorization efficiency of Orange II. In addition, $Fe^{2+}$ is readily soluble in a wide range of

pH levels (2.0–9.0), while $Fe^{3+}$ precipitates when the pH is higher than 3.0 [29]. The soluble iron catalyst might have had high availability for the activation of PMS.

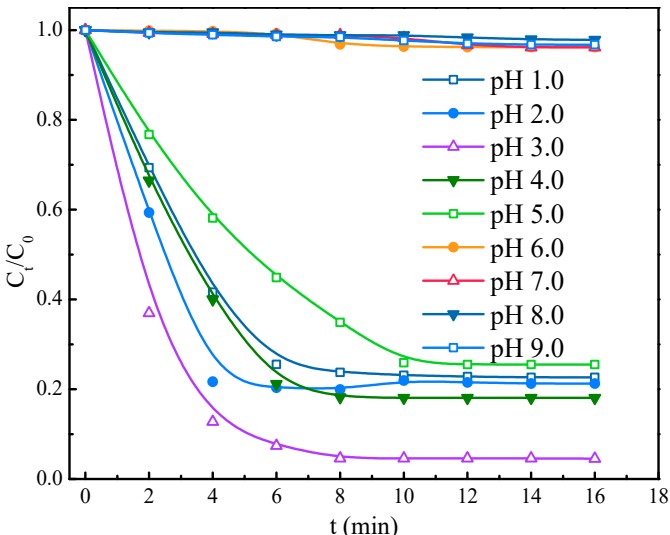

**Figure 3.** The effects of the initial pH on the decolorization of Orange II ($[Fe^{3+}]_0 = 100$ μM, [Orange II]$_0 = 50$ μM, [PMS]$_0 = 150$ μM, [oxalate]$_0 = 80$ μM, under UVA irradiation, $\lambda_{irr.} = 365$ nm).

*3.4. Effect of PMS Dosage on the Decolorization of Orange II in UVA-Fe$^{3+}$-PMS System*

The production of reactive oxidants was strongly dependent on PMS; therefore, the effect of the PMS dosage on the Orange II decolorization in the experimental system was investigated. PMS was the source of reactive oxidants, and, as shown in Figure 4, the increase in the PMS dosage from 50 μM to 200 μM led to a significant increase in the Orange II decolorization. The decolorization efficiencies of Orange II at the PMS dosages of 50, 80, 100, 120, and 150 μM were 55.9%, 75.6%, 86.2%, 92.4%, and 97.5%, respectively. This is because the production of radicals from the PMS activation increased with the increase in the PMS dose [7]; more radicals led to higher decolorization efficiency.

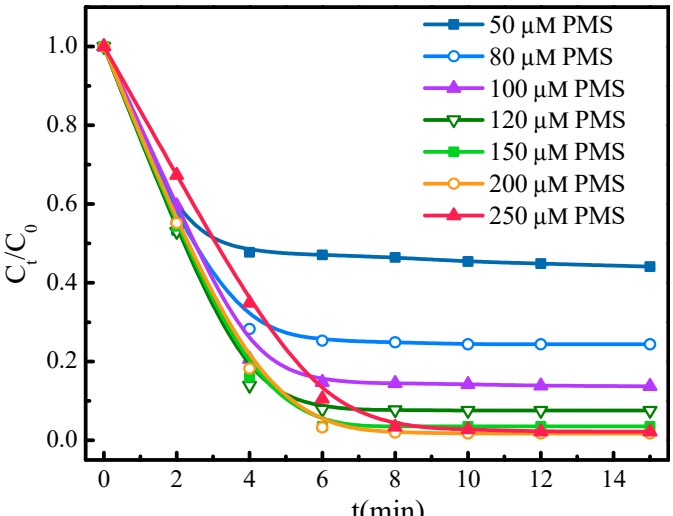

**Figure 4.** The effects of the PMS dosage on the decolorization of Orange ($[Fe^{3+}]_0 = 100$ μM [Orange II]$_0 = 50$ μM, [oxalate]$_0 = 80$ μM, under UVA irradiation, $\lambda_{irr.} = 365$ nm, pH 3).

However, a further increase in the PMS dosage did not induce an increase in the decolorization efficiency in the experimental system. Similar results have been observed in previous studies in which polychlorinated biphenyl degradation was the target con-

taminant [11,28]. As the concentrations of the $Fe^{3+}$ and oxalate were fixed, an excessive PMS dosage could hardly generate more radicals in the system. Additionally, an excessive dosage of PMS acts as a free-radical scavenger itself and therefore decreases the reactivity of radicals on Orange II [30,31]. Consequently, the optimum PMS dosage was 150 μM in the experimental system.

### 3.5. Effects of $Fe^{3+}$ and Oxalate Concentrations on the Decolorization of Orange II in UVA-$Fe^{3+}$-PMS System

The concentration of $Fe^{3+}$ plays an important role in the oxidation of organic compounds. The produced $Fe^{2+}$ via the photolysis of ferrioxalate activated the PMS to generate highly reactive sulfate radicals. In addition, both the $Fe^{3+}$-aqueous complexes and ferrioxalate could produce $^{\bullet}OH$ under UVA irradiation and may contribute to the decolorization of Orange II [32]. Therefore, the $Fe^{3+}$ concentration is a key parameter in the decolorization of Orange II in the UVA-$Fe^{3+}$-PMS-oxalate system.

As presented in Figure 5a, the decolorization efficiency of Orange II was only 21.9% in the absence of $Fe^{3+}$, while it significantly increased to 94.6% with the addition of 50 μM $Fe^{3+}$. The PMS directly oxidized the Orange II at pH 3.0, while the $k_{obs}$ was low (0.015 $min^{-1}$). All decolorization efficiencies of Orange II were above 94% with the addition of $Fe^{3+}$ (from 50 to 150 μM). The variation in the $Fe^{3+}$ concentrations in our experiments slightly affected the decolorization efficiency of Orange II, while the decolorization rate constants of Orange II increased from 0.260 $min^{-1}$ to 0.718 $min^{-1}$, with an increase in the $Fe^{3+}$ concentrations. More $Fe^{2+}$ was provided to promote the activation of PMS with increasing $Fe^{3+}$ concentrations. However, an excessive amount of $Fe^{2+}$ can react with $SO_4^{\bullet-}$ at diffusion control rates and be detrimental to decolorization efficiency. However, the $Fe^{3+}$ concentration did not affect the decolorization efficiency in our system. This may have been because more $^{\bullet}OH$ was generated through $Fe^{3+}$ photolysis, offsetting or even overweighing the quenching of $SO_4^{\bullet-}$ by excessive $Fe^{2+}$.

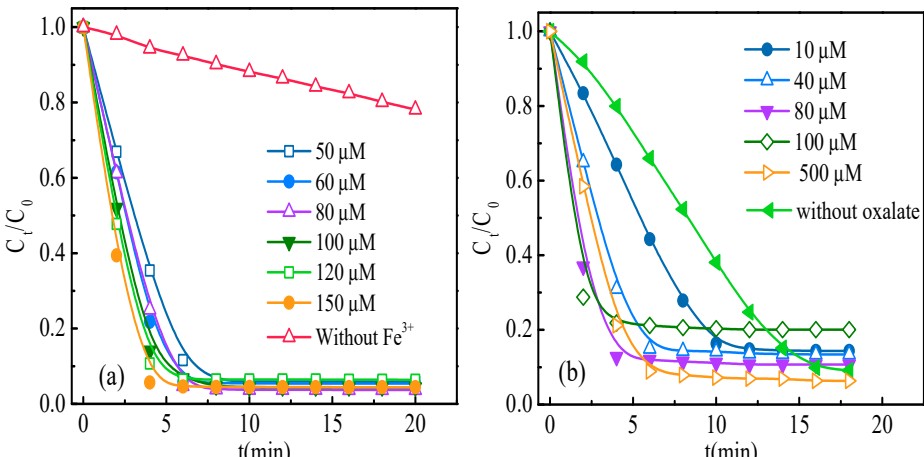

**Figure 5.** The effects of the $Fe^{3+}$ concentration (**a**) and oxalate concentration (**b**) on the decolorization of Orange ($[Fe^{3+}]_0$ = 100 μM $[Orange\ II]_0$ = 50 μM, $[PMS]_0$ = 150 μM, under UVA irradiation, $\lambda_{irr.}$ = 365 nm, pH 3).

When the $Fe^{3+}$ concentration was fixed at 100 μM, the effect of the oxalate concentration on the decolorization of Orange II was studied, as shown in Figure 5b. An increased decolorization rate of Orange II was observed from 10 μM to 80 μM oxalate, after which the increase in the oxalate concentration led to decreases in the decolorization efficiency and rate. Due to the high photoactivity of ferrioxalate, $Fe^{3+}$ can be rapidly converted into $Fe^{2+}$ under UVA irradiation. Additionally, the appropriate amount of oxalate effectively accelerated the redox cycle of $Fe^{3+}/Fe^{2+}$ and promoted the activation of the PMS. However,

a relatively high concentration of oxalate also competed with the Orange II for $SO_4^{\bullet-}$ attack and reduced the decolorization efficiency and rate of Orange II.

### 3.6. Effects of Coexisting Anions on the Decolorization of Orange II in the UVA-$Fe^{3+}$-PMS System

In AOP degradation experiments, a saline solution is commonly used as a reaction solution. To explore the potential effects of anions in the UVA-$Fe^{3+}$-PMS system, experiments on the decolorization of Orange II were also conducted in the presence of different anions ($Cl^-$, $CO_3^{2-}$, $SO_4^{2-}$, and $NO_3^-$) with different concentrations (0.01, 0.1, and 1.0 mM).

As presented in Figure 6, similar decolorization efficiencies were found under different experimental conditions with different coexisting anions at different concentrations. The $k_{obs}$ was almost not affected by the addition of 0.01 mM anions. At relatively high concentrations, the different anions exhibited different effects, $k_{obs}$ (in Table 1), in the presence of different anions, following the order of $Cl^- > NO_3^- > SO_4^{2-} > CO_3^{2-}$.

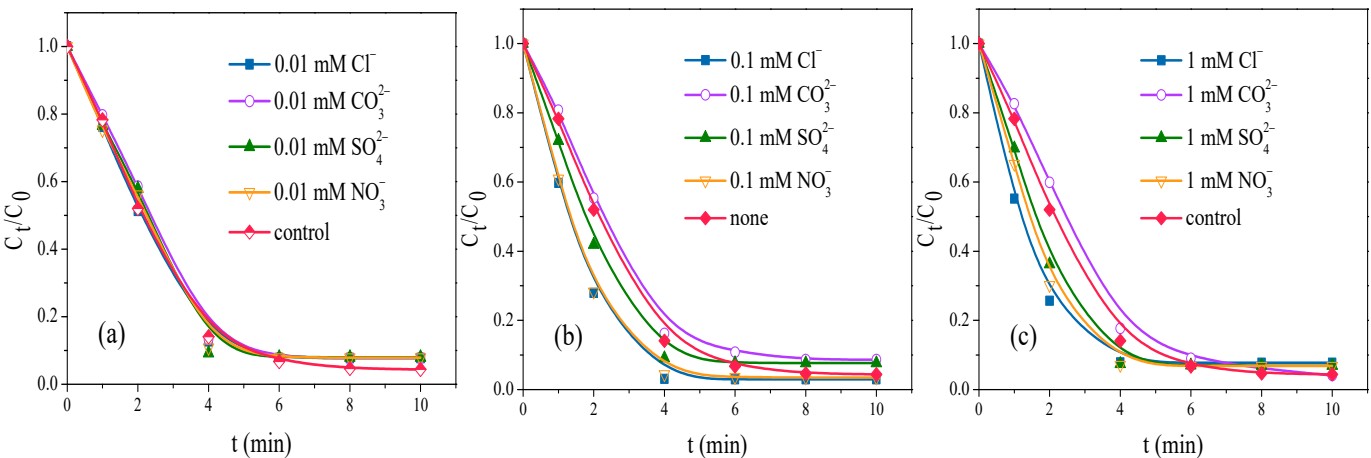

**Figure 6.** The effects of coexisting anions at 0.01 mM (**a**), 0.1 mM (**b**), and 1 mM (**c**) on the decolorization of Orange ([$Fe^{3+}$]$_0$ = 100 μM, [Orange II]$_0$ = 50 μM, [PMS]$_0$ = 150 μM, [oxalate]$_0$ = 80 μM, under UVA irradiation, $\lambda_{irr.}$ = 365 nm, pH 3).

**Table 1.** $k_{obs}$ in the UVA-$Fe^{3+}$-PMS-oxalate system in the presence of different coexisting anions at different concentrations.

| Anion | $k_{obs}$ (min$^{-1}$) (0.0 mM) | $k_{obs}$ (min$^{-1}$) (0.01 mM) | $k_{obs}$ (min$^{-1}$) (0.1 mM) | $k_{obs}$ (min$^{-1}$) (1 mM) |
|---|---|---|---|---|
| $Cl^-$ | | 0.525 | 0.679 | 0.887 |
| $NO_3^-$ | 0.515 | 0.565 | 0.668 | 0.798 |
| $SO_4^{2-}$ | | 0.601 | 0.613 | 0.642 |
| $CO_3^{2-}$ | | 0.514 | 0.467 | 0.444 |

The presence of $Cl^-$ significantly enhanced the decolorization rate of Orange II, which could be attributed to the reaction of $Cl^-$ with PMS generating chorine-containing radicals ($Cl^\bullet$, $Cl_2^{\bullet-}$, and $ClHO^{\bullet-}$) and free available chlorine, accounting for the acceleration of the Orange II decolorization [33]. Although $NO_3^-$ could scavenge $SO_4^{\bullet-}$ and $^\bullet OH$ and produce nitrate radicals with a lower redox potential of 2.3–2.7 V, it could produce $^\bullet OH$ through UV photolysis with a quantum yield of 0.09 [34]. $SO_4^{2-}$ showed a slight promoting effect on the Orange II decolorization due to the decrease in the PMS decomposition with increasing ionic strength [35]. The decolorization rate of Orange II slowed down with the addition of $CO_3^{2-}$. $CO_3^{2-}$ acts as a scavenger to $SO_4^{\bullet-}$ or $^\bullet OH$ and forms less reactive radicals, such as $CO_3^{\bullet-}$ and $HCO_3^{\bullet-}$ [36].

*3.7. Mechanisms of Decolorization of Orange II in the UVA-Fe$^{3+}$-PMS-Oxalate System*

3.7.1. Predominant Oxidants for Orange II Decolorization

In the UVA-Fe$^{3+}$-PMS-oxalate system, possible reactive species, including $SO_4^{\bullet-}$, $^{\bullet}OH$, $HO_2^{\bullet}/O_2^-$, and $^1O_2$, might be generated according to Reactions 7–12 [8]. To identify the reactive oxidants responsible for Orange II decolorization in the experimental system, a comprehensive investigation was conducted using ESR technology and radical scavengers.

$$Fe^{2+} + HSO_5^- \rightarrow Fe^{3+} + SO_4^{\bullet-} + HO^- \tag{7}$$

$$SO_4^{\bullet-} + H_2O \rightarrow H^+ + SO_4^{2-} + {}^{\bullet}OH, \tag{8}$$

$$SO_4^{\bullet-} + OH^- \rightarrow SO_4^{2-} + {}^{\bullet}OH, \tag{9}$$

$$2Fe^{3+} + 2HSO_5^- + 6H_2O \rightarrow 2Fe^{2+} + 4HO_2^{\bullet-} + 2SO_4^{2-} + 10H^+, \tag{10}$$

$$Fe^{2+} + O_2 \rightarrow O_2^{\bullet-} + Fe^{3+}, \tag{11}$$

$$2O_2^{\bullet-} + 2H_2O \rightarrow {}^1O_2 + H_2O_2 + 2OH^-, \tag{12}$$

- $SO_4^{\bullet-}$ and $^{\bullet}OH$

To confirm free-radical generation via the electron transfer process in the UVA-Fe$^{3+}$-PMS-oxalate system, the production of $SO_4^{\bullet-}$ and $^{\bullet}OH$ was investigated by ESR technology. DMPO-OH and DMPO-SO$_4$ adducts were observed in both UVA irradiation and dark conditions (Figure 7a,d). The EPR peaks had much higher intensity under the UVA irradiation than that in the dark. This indicates that the UVA irradiation significantly promoted the generation of radicals in this system. The signal of the DMPO-OH adduct was much stronger than that of the DMPO-SO$_4$ adduct, probably due to the conversion of DMPO-SO$_4$ to DMPO-OH or the insignificant production of DMPO-SO$_4$. In addition, both the intensities of the DMPO-OH and DMPO-SO$_4$ adducts obviously increased with the reaction time.

To identify which free radical acted as the primary oxidant for Orange II decolorization in the UVA-Fe$^{3+}$-PMS-oxalate system, the effects of radical scavengers (isopropyl alcohol (IPA) and tert-butyl alcohol (TBA)) on the decolorization of Orange II were investigated. IPA can react efficiently with both $SO_4^{\bullet-}$ and $^{\bullet}OH$ (($k_{(\bullet OH, IPA)}$ = 1.2–2.8 × 10$^9$ M$^{-1}$·s$^{-1}$, $k_{(SO4\bullet-, IPA)}$ = 1.6–7.7 × 10$^7$ M$^{-1}$·s$^{-1}$)), whereas TBA can selectively react with OH ($k_{(HO\bullet, TBA)}$ = (3.8–7.6) × 10$^8$ M$^{-1}$·s$^{-1}$, $k_{(SO4\bullet-, TBA)}$ = (4.0–9.1) × 10$^5$ M$^{-1}$·s$^{-1}$) [37], and thus, the discrepancy in the quenching effect could allow us to differentiate the contribution of $SO_4^{\bullet-}$ and $^{\bullet}OH$. The addition of IPA as a scavenger for both $SO_4^{\bullet-}$ and $^{\bullet}OH$ significantly reduced the degradation of Orange II, which is consistent with the results of the ESR experiment. As shown in Figure 8a, approximately 95.0% of the Orange II decolorization was inhibited in the presence of 130 mM IPA. However, the decolorization of Orange II was slightly reduced when TBA was used as the $^{\bullet}OH$ scavenger. The decolorization efficiency of Orange II decreased to 88.7% in the presence of 104 mM of TBA (Figure 8b). This result suggests that $SO_4^{\bullet-}$ rather than $^{\bullet}OH$ was the predominant oxidant that was responsible for Orange II decolorization in the UVA-Fe$^{3+}$-PMS-oxalate system, $^{\bullet}OH$ played a secondary role in the decolorization of Orange II. Furthermore, the addition of IPA did not completely suppress the Orange II decolorization, indicating that other oxidants also contributed to the decolorization of Orange II in the UVA-Fe$^{3+}$-PMS-oxalate system.

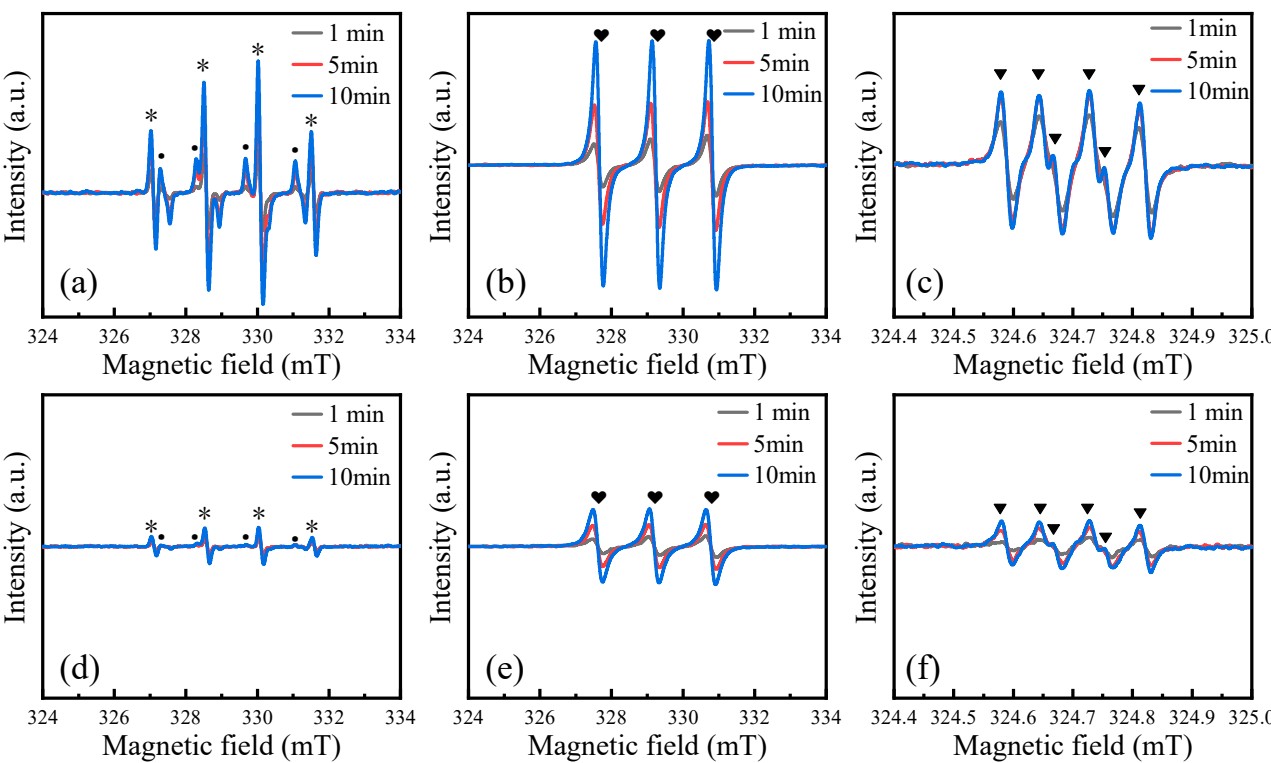

**Figure 7.** Signals of reactive oxidants in UVA-$Fe^{3+}$-PMS-oxalate system under UVA irradiation and in the dark ($SO_4^{\bullet-}$ and $^{\bullet}OH$ under UVA irradiation (**a**) and in the dark (**d**); $^1O_2$ under UVA irradiation (**b**) and in the dark (**e**); $HO_2^{\bullet}/O_2^{\bullet-}$ under UVA irradiation (**c**) and in the dark (**f**)). * signal of DMPO-OH adduct; $\bullet$ signal of DMPO-$SO_4$ adduct; $\heartsuit$ signal of DMPO-$^1O_2$ adduct; $\blacktriangledown$ signal of $HO_2^{\bullet}/O_2^{\bullet-}$ adduct.

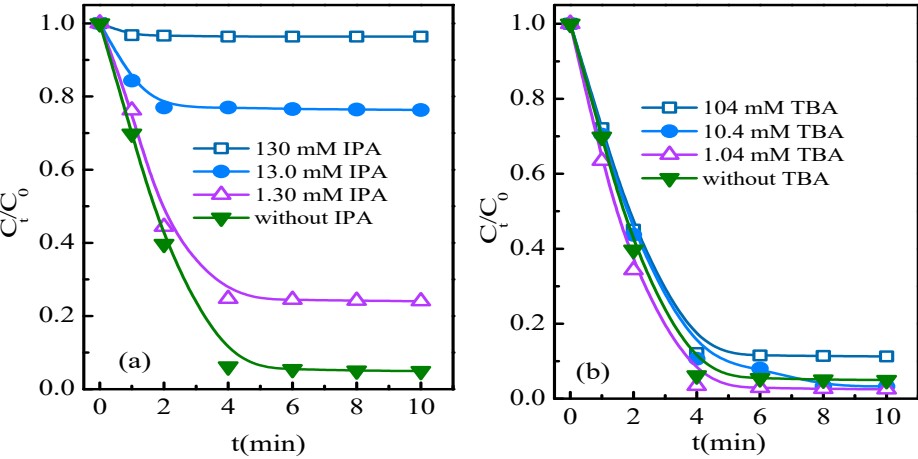

**Figure 8.** The effects of IPA (**a**) and TBA (**b**) on the decolorization of Orange ($[Fe^{3+}]_0 = 100$ μM, $[Orange\ II]_0 = 50$ μM, $[PMS]_0 = 150$ μM, $[oxalate]_0 = 80$ μM, under UVA irradiation, $\lambda_{irr.} = 365$ nm, pH 3).

- $HO_2^{\bullet}/O_2^{\bullet-}$ and $^1O_2$

In a previous work, superoxide radicals ($HO_2^{\bullet}/O_2^{\bullet-}$) and singlet oxygen ($^1O_2$) were reported to be generated during the activation of PMS [38]. ESR experiments also proved the presence of $O_2^{\bullet-}$ and $^1O_2$ in the experimental system under UVA irradiation (Figure 7b,c) and in the dark (Figure 7e,f). Therefore, the contributions of $O_2^{\bullet-}$ and $^1O_2$ to the decolorization of Orange II should be assessed.

Trichloromethane ($CHCl_3$, TCM) is an effective $HO_2^{\bullet}/O_2^{\bullet-}$ quencher that can capture electrons so as not to form $HO_2^{\bullet}/O_2^{\bullet-}$. A series of different concentrations of TCM were added to explore the contribution of $HO_2^{\bullet}/O_2^{\bullet-}$ to Orange II decolorization. Figure 9a shows the effects of TCM on the decolorization of Orange II in the experimental system. The decolorization efficiency of Orange II was partly inhibited, and it decreased to 87.7% in the presence of 124 mM TCM, which is very close to that in the presence of 104 mM TBA. Considering TCM is resistant to $^{\bullet}OH$ ($k_{(^{\bullet}OH, \, TCM)} < 2 \times 10^6 \, M^{-1} \cdot s^{-1}$,) [39], the results of the scavenging experiment by TCM clearly contradict those of TBA. Furthermore, $^{\bullet}OH$ has a stronger oxidative capacity than $HO_2^{\bullet}/O_2^{\bullet-}$. This indicates that $HO_2^{\bullet}/O_2^{\bullet-}$ might be the precursor of $^{\bullet}OH$, which indirectly participated in the decolorization of Orange II. Furthermore, the similar inhibition of TCM and TBA on the decolorization of Orange II demonstrates that $^{\bullet}OH$ formation strongly depended on $HO_2^{\bullet}/O_2^{\bullet-}$.

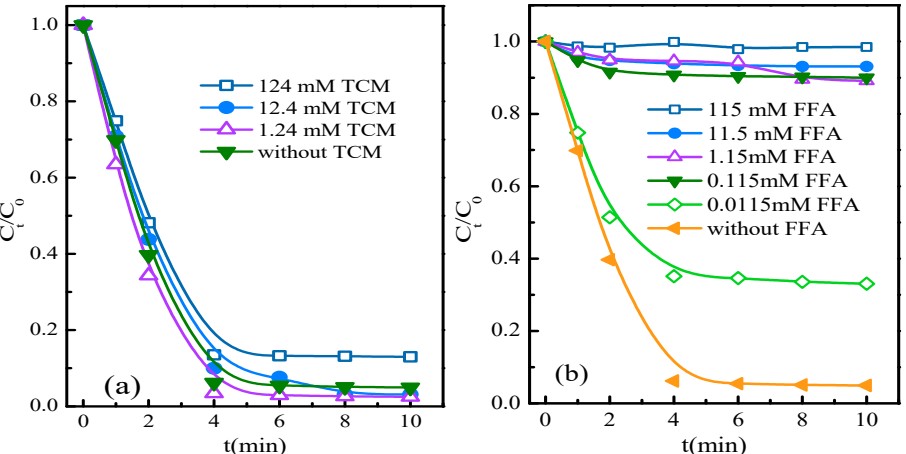

**Figure 9.** The effects of TCM (**a**) and FFA (**b**) on the decolorization of Orange ([$Fe^{3+}$]$_0$ = 100 μM, [Orange II]$_0$ = 50 μM, [PMS]$_0$ = 150 μM, [oxalate]$_0$ = 80 μM, under UVA irradiation, $\lambda_{irr.}$ = 365 nm, pH 3).

FFA was adopted as an $^1O_2$ scavenger in the UVA-$Fe^{3+}$-PMS-oxalate system. As presented in Figure 9b, the degradation of Orange II was completely halted by the addition of the FFA. Considering that FFA is able to react with $^1O_2$ and radicals, the distinction in the extent of inhibition between the IPA and FFA was due to the involvement of $^1O_2$ in the Orange II decolorization. As far as the reactivity with radicals and $^1O_2$ is concerned, Orange II is expected to react faster with radicals [1], and $^1O_2$ makes a slight contribution to the decolorization of Orange II. The generated $^1O_2$ could arise from the self-decomposition of PMS or the dissolution of $O_2$. The self-decomposition of PMS could be excluded from the experimental system because it generally takes place under neutral and alkaline conditions. Therefore, $^1O_2$ might come from the $O_2$ in the PMS system.

Conclusively, reactive oxidants coexisted in the UVA-$Fe^{3+}$-PMS-oxalate system, including $SO_4^{\bullet-}$, $^{\bullet}OH$, and $HO_2^{\bullet}/O_2^{\bullet-}$. $^1O_2$. $SO_4^{\bullet-}$ was the predominant oxidant for the decolorization of Orange II, and the other three oxidants directly or indirectly participated in the decolorization process. Specifically, 85.8% of the Orange II decolorization was induced by $SO_4^{\bullet-}$ oxidation, while 9.0% of the Orange II decolorization could be attributed to $^{\bullet}OH$ attack. The remaining 5.2% was attributed to the reaction between Orange II and $^1O_2$. Furthermore, the formation of $^{\bullet}OH$ in the experimental system strongly depended on the $HO_2^{\bullet}/O_2^{\bullet-}$.

### 3.7.2. Possible Pathway of Decolorization

In order to explore the evolution of Orange II's molecular structure characteristics in the UVA-$Fe^{3+}$-PMS-oxalate system, the UV-vis spectra of the samples as a function of reaction time are presented in Figure 10. The spectrum exhibits four characteristic peaks at

230, 310, 430, and 484 nm. The typical peaks located at 484 nm and 430 nm were ascribed to the hydrazone form and the azo form of the dye, respectively. Furthermore, the other two peaks in the ultraviolet region, located at 230 nm and 310 nm, were ascribed to the benzene and naphthalene rings of Orange II. With the process of decolorization, the weakening effect of the peaks at 310 nm and 230 nm was not obvious, which may have been due to the intermediate products of the Orange II oxidation by those reactive oxidants. Most of these were aromatic compounds containing a benzene ring structure, which can be oxidized by the reactive oxide species in the system to oxygenated organic substances and can further be oxidized to small acids and other substances [40]. The peaks at 484 nm and 430 nm decreased significantly (the absorbance decreased from 0.365 to 0.192), indicating that the reactive oxidants produced by the system were able to directly attack the azo bond of the luminescent group in Orange II, leading to the decolorization of Orange II by breaking the conjugated $\pi$ bond [41].

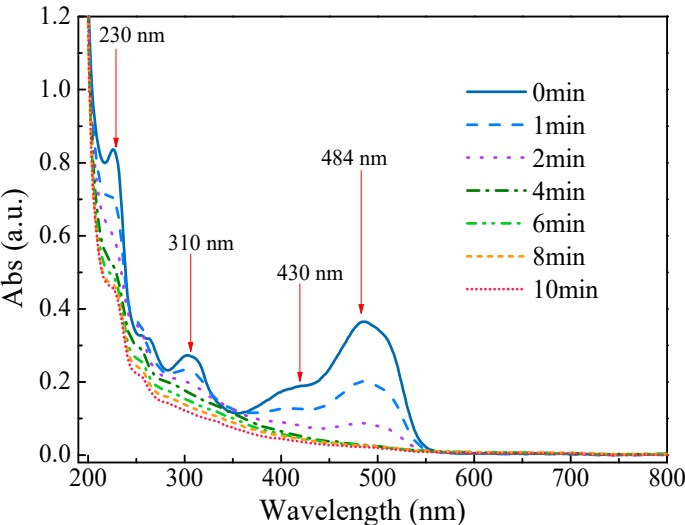

**Figure 10.** UV-vis spectra of Orange II decolorization at different times in the UVA-$Fe^{3+}$-PMS-oxalate system ([$Fe^{3+}$]$_0$ = 100 μM, [Orange II]$_0$ = 50 μM, [PMS]$_0$ = 150 μM, [oxalate]$_0$ = 80 μM, under UVA irradiation, $\lambda_{irr.}$ = 365 nm, pH = 3).

## 4. Conclusions

In this study, the performance of Orange II decolorization was investigated in the UVA-$Fe^{3+}$-PMS-oxalate system using oxalate as an accelerant. Oxalate had the greatest promoting effect on the decolorization of Orange II compared to the other six carboxylates. Efficient degradation was obtained in a pH range of 1–5, and the highest decolorization efficiency was observed at pH 3.0. Increases in the PMS dosage promoted the decolorization of Orange II. $Fe^{3+}$ concentrations employed in the experimental system had a relatively small effect on the decolorization of Orange II. The coexisting anions exhibited different effects on the decolorization process. Furthermore, the electron paramagnetic resonance and radical quenching experiments revealed that $SO_4^{\bullet-}$ was the predominant oxidant responsible for the decolorization of Orange II. $^{\bullet}OH$ and $^1O_2$ also participated in the decolorization process. The possible pathway of Orange II decolorization was presented based on the UV-vis spectrum, suggesting that the azo bond of the luminescent group in Orange II was attacked by the reactive oxidants in the experimental system. This study provides a simple and eco-friendly strategy for the treatment of wastewater containing azo dyes.

**Supplementary Materials:** The following supporting information can be downloaded at: https://www.mdpi.com/article/10.3390/pr11030903/s1, Figure S1: The absorption spectra of $Fe^{3+}$, PMS, oxalate, $Fe^{3+}$-oxalate complexes, and Orange II; Figure S2: The comparison of UVA-$Fe^{3+}$-PMS-oxalate system and $Fe^{3+}$-PMS-oxalate system; Figure S3: The effects of light wavelength on the decolorization of Orange II; Table S1: $Fe^{2+}$ production in the UVA-$Fe^{3+}$-PMS-oxalate system at pH 3.0 at different time.

**Author Contributions:** Conceptualization, Y.W. and G.M.; methodology, G.M.; formal analysis, X.D.; investigation, X.D. and C.L.; resources, Y.W. and P.C.; data curation, P.C.; writing—original draft preparation, X.D.; writing—review and editing, Y.W. and G.M.; visualization, X.D.; supervision, G.M.; project administration, G.M.; funding acquisition, Y.W. All authors have read and agreed to the published version of the manuscript.

**Funding:** This research was funded by the National Natural Science Foundation of China, grant number (21667011), the Science and Technology Fund of Guizhou Province, grant number ZK[2022]206, and the National Undergraduate Innovation and Entrepreneurship Training Program of China, grant number 202110672029.

**Data Availability Statement:** Not applicable.

**Acknowledgments:** The authors would like to thank the laboratory staff at Karst Environmental Geological Hazard Prevention Laboratory of Guizhou Minzu University for providing technical assistance for laboratory analysis.

**Conflicts of Interest:** The authors declare no conflict of interest.

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
