# Peer review of "Efficient Decolorization of Azo Dye Orange II in a UV-Fe3+-PMS-Oxalate System"

_processes, doi:10.3390/pr11030903_

Round 1
Reviewer 1 Report
In the manuscript submitted for review, the authors presented an interesting study regarding the decolorization of azo dye Orange II using a UVA-Fe3+-PMS-oxalate system. The effect of pH, initial concentrations of PMS, oxalate, Fe3+, and coexisting anions on the decolorization process were investigated.
However, the paper should be revised before it desired to be published. The detailed suggestions are listed as follows.
1) The UV-vis spectrophotometer used for determination of residual Orange is not specified.
2) A more detailed description of the conditions that the experiments were carried out is necessary (without irradiation, in the dark, at simulated sunlight irradiation)
3) The concentrations used for carboxylates (oxalates, pyruvate, propionate, succinate, glutamate, citrate, and gluconate) is not mentioned.
Author Response
We thank the reviewer for his/her helpful comments and valuable suggestions.
1) The UV-vis spectrophotometer used for determination of residual Orange is not specified.
We have added related information about the spectrophotometer.
2) A more detailed description of the conditions that the experiments were carried out is necessary (without irradiation, in the dark, at simulated sunlight irradiation)
We have added detailed information of the experimental conditions for each figure in the revised manuscript
3) The concentrations used for carboxylates (oxalates, pyruvate, propionate, succinate, glutamate, citrate, and gluconate) is not mentioned.
The concentration used for all the carboxylates is 80 µM, we added the information in the experiment conditions and in the captions of the figures.
Reviewer 2 Report
Review Report
Manuscript ID: processes-2280643
Title: Efficient decolorization of the azo dye Orange II in a UV-Fe3+-PMS-oxalate system
The target of this investigation is the decolorization of Orange II in the UVA-Fe3+-PMS-oxalate system. It has been found that Oxalate had the most excellent decolorization effect of Orange II among six others carboxylates. Concerning pH, the highest decolorization effectiveness was detected at pH 3.0. Decolorization of Orange II was promoted with the increase in PMS quantity. A small effect was observed in the case of the influence of Fe3+ concentration. It was shown that the azo bond of the luminescent group in Orange II was attacked by the reactive oxidants in the experimental system. A simple and eco-friendly strategy treatment of wastewater containing azo dyes was presented in this study.
This study is excellently conducted with relevant results, explanations, and references. I recommend that the Editorial office consider this manuscript for publication after minor revision.
Reviewer’s Suggestion
The literature foundation of equation 1 has to be cited.
Lines 135 "UV/PMS system have .." Instead have put has.
Line 216 " … availability to activation of PMS ", should be for activation.
Author Response
We thank the reviewer for his/her helpful comments and valuable suggestions.
Reviewer’s Suggestion
The literature foundation of equation 1 has to be cited.
We have added the reference for the equation 1.
Lines 135 "UV/PMS system have .." Instead have put has.
We have replaced the word “have” by “has” in the revised manuscript.
Line 216 " … availability to activation of PMS ", should be for activation.
We have replaced the word “to” by “for” in the revised manuscript.